# Microbial Growth under Limiting Conditions-Future Perspectives

**DOI:** 10.3390/microorganisms11071641

**Published:** 2023-06-23

**Authors:** Juan M. Gonzalez, Beatriz Aranda

**Affiliations:** Instituto de Recursos Naturales y Agrobiología de Sevilla, Consejo Superior de Investigaciones Científicas, IRNAS-CSIC, E-41012 Sevilla, Spain; beatriz.aranda@irnas.csic.es

**Keywords:** microbial growth, microbial diversity, growth rate, near-zero growth, physiological states, stationary phase, continuous culture

## Abstract

Microorganisms rule the functioning of our planet and each one of the individual macroscopic living creature. Nevertheless, microbial activity and growth status have always been challenging tasks to determine both in situ and in vivo. Microbial activity is generally related to growth, and the growth rate is a result of the availability of nutrients under adequate or adverse conditions faced by microbial cells in a changing environment. Most studies on microorganisms have been carried out under optimum or near-optimum growth conditions, but scarce information is available about microorganisms at slow-growing states (i.e., near-zero growth and maintenance metabolism). This study aims to better understand microorganisms under growth-limiting conditions. This is expected to provide new perspectives on the functions and relevance of the microbial world. This is because (i) microorganisms in nature frequently face conditions of severe growth limitation, (ii) microorganisms activate singular pathways (mostly genes remaining to be functionally annotated), resulting in a broad range of secondary metabolites, and (iii) the response of microorganisms to slow-growth conditions remains to be understood, including persistence strategies, gene expression, and cell differentiation both within clonal populations and due to the complexity of the environment.

## 1. Introduction

At present, it is accepted that microorganisms govern the functioning of the ecosystems and multicellular organisms on Earth [1,2,3,4,5]. The biogeochemical cycles of elements are regulated by the activity of microorganisms and a number of critical steps are only carried out by microorganisms (e.g., numerous processes involving inorganic nitrogen transformation, methane production and oxidation, oxidation of metal sulfides, iron oxidation and reduction, etc.) [6,7]. This has important consequences for the outcomes of global processes, such as climate change, because of the role of microorganisms in maintaining carbon equilibrium with the atmosphere [3,8,9,10,11]. In addition, in the last years, recent research has shown that microorganisms rule critical aspects of plant and animal (including humans) physiology [4,6,12,13]. Even psychological aspects of human behavior have been demonstrated to be dependent on the microbiome of each individual person [14,15].

Microorganisms play critical roles in our planet’s sustainability and maintenance, but how this regulation occurs and under which conditions specific microorganisms carry out expected functions require further investigation [1,16,17,18]. To this aim, the high microbial abundance and diversity existing in nature contribute to generating highly complex systems. For instance, soils have been reported to contain about 10^10^ bacteria g^−1^ [19,20] with around 30,000 different microbial taxa [19,21,22] and clean water might contain around 10^6^ bacteria mL^−1^ with potentially thousands of different taxa [19,23]. Obviously, different microorganisms are represented by specific proportions of cells within communities [18,19,23,24] and their abundance and activity vary over time depending on changes in the environment. These changes are the result of dynamic physical (e.g., temperature, humidity, etc.), chemical (availability of organic and inorganic nutrients, pH, inhibition by pollutants, etc.), and biological (competition, predation, diversity, etc.) factors that are involved in the natural environment [25]. At present, a major challenge in microbiology is to be able to evaluate and understand the growth status and activity of microbial communities and the contributions of their different microbial components. This study is mainly focused on prokaryotes (bacteria and archaea), but it can also be applied to some other microorganisms such as fungi (i.e., yeast) and protists.

Growth is a result of microbial activity [1,26], but quantification of the growth status of a microbial population or an individual cell is an issue that requires further investigation and remains a major challenge to be solved. Bacterial growth has been frequently assessed in laboratory clonal cultures under optimal conditions [27]. Most models and studies have been based on microorganisms growing under these conditions. Nevertheless, microorganisms in nature are rarely under such ideal conditions. Because nature is a highly competitive environment, microbial growth is generally severely limited and strictly regulated [27] so than microbial abundance and biomass are in equilibrium within the carrying capacity of the ecosystem and the planet [17]. Consequently, microbial growth and functionality are time-dependent environmental conditions and we are barely starting to understand how biotic and abiotic factors influence the growth of microorganisms [28,29,30,31].

It is believed that microorganisms in the natural environment thrive through feast and famine cycles [26,29,30,31,32,33] due to environmental conditions and the availability of nutrients. While short pulses of nutrient abundance can occur in nature, life in the environment is assumed to mostly represent famine conditions with nutrients strictly limiting microbial growth. Although nutrients are a major factor limiting or regulating growth, numerous biotic and abiotic factors can exert stress situations that result in the activation or inhibition of microbial growth, showing differential effects among distinct microbial species. Biotic factors could include, for example, competition with other microorganisms within the same species (i.e., sharing available nutrients) or with other species (i.e., inter-species competition) over the same nutrient sources, which is a consequence of the microbial diversity at the studied ecosystem. Grazing (i.e., protistan grazers) or viral infection prevent or limit growth and have major effects on limiting the abundance of specific microbial species. Among the abiotic factors that directly limit microbial growth, one of the most drastic factors is temperature. Low temperatures might result in a highly significant reduction in growth rate, while high temperatures can also inhibit the growth of mesophiles and, for example, activate the growth of thermophiles. Each microbial taxon has a specific optimum growth temperature. As previously observed [34], microorganisms can grow within a narrower range of temperatures in nature than in the laboratory. This suggests increased potential competition among microbial species outcompeting poorly fit taxa when diverging from their optimal temperatures. There is a long list of other abiotic factors that can affect microbial growth, such as pH, water content, the presence of pollutants or growth inhibitors, etc. [35,36,37,38]. Each one of these factors, or a sum of several of them, can drastically influence growth rates.

Microorganisms are able to adapt to a very broad range of environmental conditions [28,39], reflecting the huge microbial diversity on Earth [19,28]. Microorganisms can be found practically everywhere on our planet, but every microorganism is not able to thrive everywhere [28,40]. The environmental conditions are believed to restrict microbial distribution [28,40]. Clear examples are found among the microorganisms from extreme environments (i.e., extremophiles) [40,41,42]. For example, high-temperature environments will not allow the growth of standard mesophiles and will cancel all possibility of their survival [40]; however, hyperthermophiles must grow in high-temperature environments. Similarly, halophiles inhabit high-salt environments [41] but will not prosper in freshwater environments, while non-halophiles will not survive in salterns.

In spite of the restrictive growth conditions in the environment, microorganisms are generally believed to persist through a variety of adverse conditions and regrow when appropriate conditions reappear [39,43,44,45].

Microbial communities are a composite of a large number of microbial taxa [19,23,39]. However, all of those taxa are not represented in equal numbers. Rather, some species are highly abundant and others are rare or very poorly represented, although they can be ubiquitous [19,23,24,45]. As a result, microbial growth in the environment must be considered as a net sum of growth and decay for each species integrated in the community. The fact that some microbial species grow at a relatively high rate and others grow poorly at a near-zero growth rate or even experience decay (reduction of abundance) implies that the environment is providing adequate growth conditions for some and adverse or limiting conditions for others. In addition, different taxa can have different growth capabilities, so that under optimum conditions some microorganisms are fast growers and others always show slow growth rates [19,23]. This is a variable over time and the depletion of specific nutrients or changes in abiotic or biotic factors will result in changes in the growth rates of components of microbial communities. Thus, microbial growth is a time-dependent variable within highly dynamic microbial communities [46,47]. Potential adaptations and changes in the communities occur in part due to high genome plasticity and in part due to the great adaptability of microorganisms [29,30,31,32,48,49,50]. The time dependency and time series of microbial community composition and growth need to be studied [45,46,47,49,50,51] and research attention is required to fully understand the relevance of microorganisms in nature.

As microbial diversity is huge, the mechanisms of responses and growth rates for different microorganisms under a broad range of conditions are expected to be similarly diverse. Molecular biology analysis, including genomic and transcriptomic studies, are techniques that will be required to approach these types of issues. Some examples of the mechanisms responsible for optimum and limited growth have been already reported in the literature, mainly for specific model bacteria in periods of nutrient depletion. Those responses have been studied mostly related to the development of the stationary phase of growth (and not directly related to the scenario of minimum growth rates; see below for the different stationary phases and slow growth concepts) and were reported to be controlled by master regulators [52,53,54] including, as some examples, alternative sigma factors such as RpoS and RpoH [55,56], small molecule effectors such as ppGpp [57], gene repressors such as LexA [57], and inorganic molecules, among other responses to stress and adversity including down- and upregulation of specific genes and processes such as GASP (Growth Advantage in Stationary Phase) phenotypes [52,58,59,60], SCDI (Stationary phase Contact-Dependent Inhibition) phenomena [61,62], DNA polymerases [56], error-correcting enzymes [60], movement of mobile genetic elements [49,50], metabolic slowdown [63,64,65], etc. An interesting aspect is that microorganisms under limiting or adverse conditions have been reported to activate metabolic machinery to generate secondary metabolites [66], which are not usually required for growth but can be beneficial in periods of adversity to outcompete other species or protection during periods of persistence and maintenance [66,67,68,69,70]. Some secondary metabolites have been identified and typical examples include antimicrobials or vitamins [66,67,68,69,70]. It is expected that by understanding bacterial behavior under growth-limiting conditions scientists could discover novel genes and their metabolic pathways which, at present, remain to be annotated.

Current gaps in our understanding of microbial growth and activity in the environment [17,18,22] limit the potential uses of microorganisms and the appropriate management of ecosystems due to lack of knowledge about the microbial world. These gaps are expected to be filled in the future so that scientists can ideally understand and manage natural ecosystems and fully comprehend their roles, implications, and the dynamic nature of the environments [18,22,29]. This study aims to highlight the interest in and relevance of studying microbial behavior under growth-limiting conditions as the next, and necessary, step towards better understanding microbial life and its role in the environment.

## 2. Microbial Growth

Microbial growth has been intensively analyzed and modeled using bacterial monospecific cultures in the laboratory [71,72,73]. As a consequence, the expected growth of a bacterial isolate in the laboratory is represented by a sigmoidal curve composed of differentiated and characteristic growth phases (Figure 1): the lag phase of growth (initiation of the metabolic machinery for growth), the exponential phase of growth (exponential increase in cell abundance and maximum growth rate), the stationary phase of growth (cells adapt to changing conditions due to scarcity of nutrients), and the decline (or death) phase (decrease in cell number).

Different mathematical models have attempted to improve the fitting of growth curve data [72,73,74,75,76]. A typical growth curve is generally represented, for instance, by the Gompertz and logistic models [72,73,74,75,76], among other non-linear growth models. Thus, the bacterial growth curve for a clonal laboratory strain is considered to be well characterized. The difficulty starts when more complex systems (bacterial consortia, environmental samples, etc.) are evaluated for global growth in the environment or specifically for each of the numerous components of microbial communities. At this point there is a need for novel and efficient strategies to monitor bacterial growth rates.

The growth of bacteria in the laboratory appears to be a trivial matter. This is not always the case because scientists have been unable to grow a large number of different microbial taxa from natural environments [77,78]. This has been frequently quantified at around 1% of total bacteria [78] although this fraction is clearly dependent on the environment being studied and the range of culturing techniques that one is willing to assay. The fact is that a high fraction of microorganisms in the natural environment remains uncultured. Additionally, some microorganisms grow erratically and their growth is not easily predictable. Obviously, these scenarios are typical examples suggesting that new knowledge about microbial metabolism and growth conditions remains to be gathered. Examples of recent efforts to culture novel bacteria include the increased numbers of previously candidate phyla that scientists are starting to populate with cultured bacterial isolates. A couple of examples of these new phyla are Armatimonadetes (previously OP10) [79] and Saccharibacteria (previously TM7) [80].

The standard growth curve defines how microorganisms grow in a laboratory culture generally close to its optimum growth conditions. Although working in the laboratory one can design experiments to be able to measure a specific growth rate, this needs to be carefully evaluated because we do not know the growth rate of bacterial cells during the whole growth curve. Two points need to be mentioned. First, we are unable to experimentally calculate the growth rate of microorganisms growing in a culture at the time of one’s choice. We could, of course, make a rough estimate of the global growth rate based on the use of well-fitting mathematical growth models as the derivative in t (time) of the mathematical function being applied.

Second, at present we are not able to easily analyze single-cell parameters. There have been attempts to study single-cell genomics in prokaryotes [81] and this is currently under development, becoming increasingly feasible and accurate in the last years. Our knowledge has a gap in understanding the variability in phenotypes, metabolism, epigenetics, and environmental responses on a per cell basis in a monospecific culture or in a microbial community in the environment [81,82]. Generally, we should assume that cells in a culture, above all during transitions between growth phases (i.e., from lag phase to exponential phase, or from exponential phase to stationary phase, etc.), can adopt a wide range of possibilities of growth rates (Table 1) from zero growth to the maximum achievable growth under the provided culturing conditions.

Unfortunately, experimental quantification of the growth rate along a typical growth curve in batch can only be determined with certainty at the inflection point of the sigmoidal curve during the exponential phase of growth (Figure 1). Beyond this point, one must assume the presence of cells at a broad variety of physiological stages. The variability existing at different phases of growth remains to be studied. Some specific experimental strategies have been designed to partially solve the issue of obtaining cells at steady-state for specific growth rates. The solutions are related to the use of continuous culturing techniques (see below).

The challenge in determining growth in a natural environmental microbial community and the growth rate of specific members of that community remains to be efficiently solved. An additional complexity level arises when these determinations should be performed, ideally frequently, over time [29,47,51] in order to analyze changes in the communities, their growth rates, and the effects of environmental factors along different time scales. It is important to highlight that slow growth rate is used herein as a relative term as a function of each microorganism’s optimum (maximum) growth rate since some are fast growers but others are much slower growing cells even under their optimum growth conditions.

A potential solution to obtain cells at different growth rates in the laboratory comes from the use of continuous culturing strategies [83]. In this case, cells are grown and the culture is stabilized at a constant growth rate (i.e., steady state) governed by the dilution rate (input of fresh medium in equilibrium with the output of grown culture) in the culturing vessel [84]. This growth rate is calculated from the dilution rate (volume of fresh medium input per unit time divided by the total volume of the culture vessel) [85,86]. The growth rate in a continuous culture does not depend on other factors such as nutrient concentration in the medium. Varying the culture medium concentration will increase or reduce cell abundance in the culture but the growth rate of the cells should remain constant.

Continuous cultures can be achieved in a chemostat [85,86] but the range of growth rates must also be within certain limits. For example, the dilution rate must be lower than the maximum growth rate of the microbial strain under the provided culturing conditions, otherwise we will progressively dilute the cells out of the culture. A second restriction is that cells cannot be grown at too low growth rates because of the minimal volume of fresh medium that can continuously be pumped into the culture and homogeneously distributed [86,87].

A question arises about how to achieve even lower growth rates, those that have been named near-zero growth, close to a metabolism of maintenance, with cells growing at minimum (near-zero) rates [65,87,88]. Microbial growth at very reduced growth rates (near-zero growth) is a scarcely studied field in microbiology. The solution to studying these minimum growth conditions comes from the use of a modification of the chemostat, named a retentostat. In the retentostat, a constant volume of fresh medium is input into the culture vessel, but the whole culture is not simultaneously discarded. Instead, the culture is filtered, the cells are returned to the culture vessel, and only the used medium is discarded [87,88,89,90]. This strategy leads to a progressive accumulation of cells in the culture vessel so that a higher number of cells need to share the same amount of nutrients. As a consequence, cells must compete for the available nutrients, which leads to a progressive decrease in their growth rate. Evaluating the culture biomass over time, one can calculate the growth rate of cells at a given time point in the culture [84,85]. Following this method, a cells growth doubling time of over a year could be obtained for bacteria that doubled approximately every 20 min under optimum conditions. In nature, a typical example of bacterial cells growing at a very low growth rate is deep-subsurface bacteria, which are believed to show doubling times in the range of years to centuries [91,92,93], and their metabolism and living strategies remain mostly to be understood.

At present, monitoring methods are highly limited for rapid and efficient assessment of microbial growth rates in the environment. Methodology is needed in order to determine the growth status and/or growth rate of specific bacterial taxa in complex environmental communities. Isotopic-labeling techniques measuring the incorporation of specific labeled substrates during a relatively short incubation period are among the possibilities for estimating the bacterial activity of specific bacteria and their metabolism [94,95]. Molecular methods targeting DNA and/or RNA are another option. Molecular methods involving PCR amplification (either end-point PCR or quantitative real-time PCR) provide opportunities to develop methods for these determinations. Another alternative for molecular studies is the use of fluorescently labeled nucleotide probes so that FISH (Fluorescent In Situ Hybridization) [96] can be performed to enumerate individual cells in environmental samples and eventually classify them at different growth stages. These methods are time-consuming and the information that they provide is relatively limited. At present, only a very few model species or bacterial groups have been studied [96]. Additional perspectives have been provided using NanoSIMS [82,95], which could add relevant information for single-cell functional studies on natural samples. Methods that provide important information through rapid tests and measurements will be greatly welcomed in the years to come, enabling most of the issues mentioned herein to start being solved.

## 3. Growth at Optimum (Maximum) Rate

The most simplified scenario is that of batch culturing a well-known bacterial isolate/strain in the laboratory under optimal conditions of culture medium and physicochemical parameters (temperature, pH, salinity, available nutrients, etc.). With the standard facilities of a common laboratory, this can be easily achieved using a culturable bacterial strain. We would know, or we would be able to determine, the optimum (maximum) growth rate for that bacterial strain in a laboratory under ideal conditions. At the inflection point of the exponential phase of growth (Figure 1), cells grow at maximum (optimum) growth. Thus, one would be able to analyze the behavior (i.e., whole cell gene expression) or characteristics shown by those cells at that specific (optimum) growth rate (Table 1). This is valuable information that is frequently used to begin understanding the bacterial genome and to pursue (if needed) a taxonomic classification of the species through a polyphasic identification approach [97].

Growing bacteria in the laboratory under optimal conditions is the approach that microbiology has used mostly since its beginning [98]. Nevertheless, microorganisms in nature are rarely under optimum conditions. Different studies illustrate that microorganisms generally thrive under growth-limiting, adverse, and/or suboptimal conditions [30,31,32,33,34,92,93].

Bacteria thrive in a changing environment and their range of responses, behaviors, and alternative metabolic or physiologic strategies of survival and persistence remain to be understood [99,100]. By studying a bacterium under optimum growth conditions, our knowledge about that bacterium’s capabilities result in a quite limited set of information. Again, analyzing the functional and genomic variability for a clonal bacterial strain or for a bacterial species remains to be explored in detail [99,100,101]. How would these bacteria respond to a change in temperature, pH, salinity, or to nutrient depletion? An innumerable set of questions can be raised. A solution could start by exploring other scenarios, conditions, and, specifically, growth rates.

## 4. Growth at Slow (Limited) Rate

In the environment, microorganisms thrive somehow under limiting conditions. The factors that limit their growth is a matter to be studied and is not an easy task to solve in many cases. Microorganisms in nature that are growth-limited show rates ranging from decline to near laboratory-optimum growth rates. The final question to be raised is about the growth rate of a bacterial population growing in the environment at a given time and under specific conditions (biotic and abiotic factors) [26,30,31].

Above we have mentioned the requirement for simple methodologies to assess microbial growth rates both in the environment and in culture. This is because a major current issue is how to monitor bacterial communities using simple and rapid technology. Although some methodologies are available (e.g., single-cell genomics, NanoSIMS, FISH, etc.), these methods are too time-consuming or costly (in personnel, labor, and/or economically) to be performed frequently and to carry out calibration and experimental setups.

A way around this is to prepare cells at previously specified growth rates by using continuous cultures, using a chemostat setup [84,85,86,102]. In this way, obtaining microbial cells at steady-state growth at a defined rate is relatively straightforward using current methodology (Figure 2). The theory behind continuous culturing systems has been well established [84,85,86,102] and the growth rate is directly determined by the dilution rate [86]. The range of growth rates that can be worked out using this type of setup ranges from the optimum bacterial growth rate down to no lower than (and depending on the setup design) around 0.025 h^−1^. This rate has been reported by different researchers [85,86,87,102]. Some bacteria can show doubling times in the order of 20–30 min (corresponding to growth rates of around 1.4–2.3 h^−1^), which is the case for many laboratory model copiotrophic bacterial species (i.e., *Pseudomonas*, *Escherichia*, *Bacillus*, and many others). For these bacteria, this setup would allow bacteria to be obtained at around 100-fold lower growth rates than optimum growth rate values (above 1% of maximum growth rate). Although bacteria can thrive under scenarios of much lower growth rates (see below), this methodology allows researchers to (i) understand bacterial responses over an acceptable range of growth rates, and (ii) approach growth-limiting conditions.

Under moderate growth-limiting conditions generally achieved in a chemostat by nutrient limitation, one could start comparatively analyzing the variability in responses, preferred/unpreferred substrate temporal shifts [31,68,103], whole-gene expression (by transcriptomics or RNA sequencing) [65,104,105], physiological capabilities [92,105,106], and, likely, the production of some pathways generally repressed under optimum growth conditions [68]. Thus, the spectrum of possibilities to better understand microbial behavior and responses to the environment is opened to the researcher. This still remains a pretty much unexplored field in microbiology.

Attention should be given to differentiate bacteria showing slow growth (at limiting growth rates) from those at the stationary phase of growth [86,106]. Bacteria with slow growth are bacteria actually growing under nutrient-limiting conditions that limit their growth rate. These are growing bacteria showing steady-state growth. Bacteria at the stationary phase of growth are undergoing a process of adaptation from growing conditions to maintenance or adverse conditions so that they can persist as long as possible; this is because nutrients in their medium are depleted and conditions have started to inhibit growth [52,54,103]. The behaviors of both types of cells are clearly different and studies on cell differentiation at these two growth states could be an interesting way to better understand microbial behavior and the range of capabilities that their genomes provide to respond to different environmental scenarios. These types of analyses are needed by applying novel methodologies and combining genomic and phenotypic approaches. Whole-cell gene expression, epigenetics, physiological and metabolic responses, etc., are among the very interesting aspects to be comparatively studied.

## 5. Growth at Near-Zero Rate (Severe Growth Limitation)

In the environment, most microorganisms are under severely limited growth conditions [32,51]. This growth restriction can be the result of both biotic and abiotic factors (see above). The capability of microorganisms to persist in the environment in spite of frequent periods of adversity is a very important strategy to preserve microbial diversity and the conservancy of microbial species through Earth’s history. When nutrients are scarce or conditions are growth-restrictive, a number of microorganisms (many are still to be tested) can thrive by drastically reducing their growth to near-zero rates. Microorganisms cultured at a near-zero growth rate are viable and active cells [87,88,89,90,104,107] but the environmental limitations restrict their growth to a very slow rate. These cells at a near-zero growth rate are at a very different state than cells undergoing survival or decline (death) or at the stationary phase, as these latter processes occur when cells are no longer growing, they might show poor or null activity, and they (or a large fraction of the cells) are likely to be non-viable and/or non-culturable [108]. The metabolism of near-zero growth microorganisms is closed to that of maintenance. Cells growing at near-zero rates mostly maintain their cellular machinery and use their scarce excess energy and synthesis of organic matter to increase their biomass at the rate allowed by their supply of nutrients and their surrounding environment.

Although detailed studies are missing for most microbial species and environments, there are some examples that can explain the long-term persistence of bacteria in the environment. Cells can be at a low growth stage for really long periods, although there are no specific studies on the persistence of cells at this stage. One could hypothesize that cells could be growing at near-zero growth forever, awaiting for better conditions to arrive.

One example of cells thriving at near-zero growth rates is the persistence of viable and active cells of soil thermophiles [109,110,111,112] in cold and temperate soils. Similarly, thermophiles have been reported in arctic sediments [43,44]. The most representative bacterial group of soil thermophiles is the genus *Geobacillus* (or related Firmicutes). These cells grow optimally at around 60 °C but they can persist in temperate soils waiting for appropriate conditions to arrive. They have reached almost every place on Earth and it is assumed that they have been dispersed, for example, by winds from deserts and arid zones [113]. The activity of these cells can be easily detected by heating those soils and measuring their extracellular enzyme activity or by evaluating their RNA through quantitative RT-PCR using specific oligonucleotide probes [16,107,108]. These soil thermophiles have been found from low to high latitudes. Their activities have been demonstrated to be significant in all soils tested [109,110,114] and they are able to present a fast response to temperature increases [109,110].

Only a few tested species of microorganisms are able to grow at near-zero rates. One would be tempted to confirm that this is a general strategy among microorganisms, but this needs to be experimentally confirmed. Examples of microbial species already tested through their growth in a retentostat are *Saccharomyces cerevisiae* [87], *Lactococcus lactis* [89,90], *Bacillus subtilis* [88], *Pseudomonas putida* [115], *Nitrosomonas europaea* [104], anammox bacteria (“Candidatus *Brocadia sinica*”) [107], and *Geobacillus thermoglucosidasius* (authors’ unpublished results).

Severe growth restriction, for example (but not exclusively), as a result of nutrient limitation is a common situation in the natural environment. Microorganisms persist with minimal requirements until adequate conditions for faster growth are available in the ecosystem. While this can be a long-term strategy for different species (i.e., soil thermophiles), other species might thrive through much shorter cycles of feast and famine as a function of nutrient availability in the environment. At present, assigning microbial species and environmental scenarios to slow or near-zero growth and stationary phase-like responses remains to be studied. This will contribute to a better understanding of microbial behavior as a function of the environment.

Cells growing at near-zero rates can be obtained in the laboratory through continuous cultures using a retentostat system with biomass retention [87,88,89,90,104,107], but these low growth rates can not be obtained in batch systems. A retentostat is a closed system for cells (with complete biomass retention) but open for medium (nutrients) supply. In a retentostat, cells progressively accumulate while sharing a constant nutrient supply. Thus, cells are forced to slowly decrease their growth rates (Figure 3). Achieving near-zero growth rates can take a relatively long time, and around one or two months can be needed to obtain cells at near-zero growth rates around 10^5^-fold below their optimum (maximum) growth rate. For a fast growing bacteria (i.e., 30 min doubling time at optimum growth), this would correspond to a growth rate below 10^−4^ h^−1^, which is greater than 1000-fold below the proposed limit for a standard chemostat culturing system (see above). Longer experimental runs of the retentostat culturing system could lead to even lower growth rates. The process should allow researchers to approach those unimaginable slow growth rates that have been proposed, for instance, for microorganisms inhabiting the deep-subsurface environment [1,90,91,92,93,96]. Those deep-subsurface microorganisms have been proposed to be thriving at doubling times in the order of hundreds of years (corresponding to rates at or below 10^−6^–10^−7^ h^−1^). Some other scenarios such as polar environments are examples of microorganisms thriving at extremely low growth rates. This strategy might represent an experimental setup able to assess the living of these bacteria in the laboratory.

Retentostat culturing systems are an alternative for studying bacteria at near-zero growth, but these experimental setups can face some experimental issues that need to be solved for each microbial species to be grown. These long-lasting cultures are prone to potential contamination, and so different strategies can be proposed for specific species. One example is the use of antibiotic-resistant strains of the species to be studied. Another alternative is using selective culture media that could inhibit the growth of most potential contaminating microorganisms. In addition, the filtering of cells back to the culture vessel during long-term runs could potentially generate obstructions. Cells growing in a retentostat should maintain their viability and be actively growing. Different procedures to check for active microbial cells or to evaluate live/dead cells have been proposed for retentostat runs [87,88,89,90]. Other potential issues in these retentostat culturing systems of specific microorganisms can come up during long runs, representing additional difficulties for carrying out research on bacteria at near-zero growth.

Cells at near-zero growth are a new area of microbiology that remains scarcely explored. At near-zero growth, most metabolic activity is assumed to be directed to cell maintenance [65,67]. Cells require a minimum metabolism to maintain their cellular machinery at an active state, which basically represents the minimum requirement to keep a cell viable and able to actively grow much faster if the right conditions become present. Cells at near-zero growth represent a potential approach to better understanding the maintenance metabolism and long-term persistence of microorganisms [26,67,68,107].

Microbes living at minimum growth rates is not an unusual event and reports state that microorganisms show mechanisms of response under nutrient-limited conditions that are otherwise repressed in the cells. Among the nutrient deprivation-activated pathways are those that produce secondary metabolites. The metabolites are not required for normal growth, but when growth conditions become highly restrictive some secondary metabolites are produced [26,27,66,67]. These compounds frequently involve generally unknown metabolic pathways that remain to be annotated in the genome. Examples of secondary metabolites include a long list of antimicrobials and vitamins [69,70] required to outcompete other species, to compensate for nutrient deficiencies, and to foster growth or survival. These molecules and their functions under nutrient-deprived conditions are essential to understanding microbial growth, cell-to-cell interactions, maintenance, and survival.

Efficient microbial persistence in the environment requires different mechanisms [52,53,54]. This is achieved using low nutrient consumption strategies developing minimum requirements with minimum loses [67,68,104,107]. The development and adaptations for carrying out minimum metabolism (i.e., near-zero growth and maintenance metabolism) is the best option to support long-term persistence in the environment [67,104,107,115]. Microbial behavior under nutrient deficiency shows types of responses that result in near-zero growth, which explains the persistence of microbial diversity through adverse and nutrient-limiting conditions in the environment. In addition, facing environmental stresses such as nutrient deprivation induces potential changes both in the genomes of these cells and differential gene expression among cells with potentially critical consequences, generating genetic and epigenetic differentiation that remains to be studied. This can lead to the generation of new diversity in the microbial world. The expected rates of adaptation and mutation are assumed to be much faster under nutrient-limiting conditions or under stress than under optimum growth conditions. Thus, future research should confirm if microbial behavior under stressed conditions (i.e., nutrient depletion) enhances the persistence and survival abilities of microorganisms generating diversification in microbial populations.

Near-zero growth represents a biotechnological model of interest because this growth scenario presents microbial biomass that can be kept nearly constant through time, leading to maximum productivity of specific metabolites and increased efficiency. In this manner, the production of specific metabolites or products could be easily maximized at reduced costs [115].

Some of the differential characteristics of the distinctive growing states of microorganisms are shown in Table 1, highlighting different aspects of functional interest in microorganisms at different growth rates, their relevance to understanding processes, their activity in microbial communities, as well as the potential effects of different conditions on microbial cells.

## 6. Non-Growing Stages

Bacteria can also be found in the environment at non-growing stages. Although this review is about growth, one cannot forget that microorganisms can also thrive at non-growing stages during their life cycle. These periods can present negative growth (that is, decay of cell number or biomass) or zero net growth (equilibrium between decay and growth). Typical non-growth stages are the stationary phase of growth, the decline (death) phase of the growth curve, decay or survival periods generated by a variety of factors (antimicrobials, starvation, adverse factors, etc.), and the late stages of the stationary phase.

### 6.1. Stationary Phase of Growth

In a batch culture, after the exponential phase of growth, nutrients become scarce and the generated by-products can initiate a period that forces bacteria to adapt to the changing conditions in order to survive the non-ideal conditions generated in the culture (Figure 1) [99,100,101]. This is a period that generally is represented by zero net growth. Some cells might be dying while others might take advantage of the released nutrients as result of cell lysis to keep growing at a limited rate [58,59,103,116]. As the stationary phase of growth progresses, conditions become increasingly adverse and so growth becomes more limited over time. Cell death might increase, leading to the decline or decay (death) phase of growth (Figure 1). The result is a decline phase (see below) and late stages of the stationary phase. The ultimate goal of changes in the microbial population is to maximize survival [56,59,71].

A number of mechanisms occurring during the stationary phase of growth have been reported and these are increasingly relevant at the late stationary phase (Figure 1). A few of them are mentioned below, although further investigation will be needed to fully understand the physiology, gene expression, and evolutionary advantages of different phenotypes. The application of this information will be relevant to fields such as biotechnology and the understanding of microbiomes, among a variety of disciplines for which this information will be highly useful.

Among the reported mechanisms specific of the stationary phase of growth, the existence of stationary phase promoters (with no activity during the exponential phase) [54] presents novel perspectives for understanding cells at this phase. In addition, they present interesting potential for biotechnology and large-scale protein production [54]. At this stage, the cells are oriented to survive the adverse changes in the consumed culture media. With increasing accumulation of toxic by-products, the cells reprogram their cellular machinery to adapt to these new conditions [52,53,54], avoiding growth by activating master regulator networks [52,53,54]. A critical key component is the activation of ppGpp and pppGpp [57], which diverts the cells to consuming their own biomass resources, such as reserves and amino acids, to extend survival time. The presence of ppGpp and its derivatives act as regulators that activate some of the required processes. One of them is alternative sigma factor RpoS [55,56]. RpoS is selectively used during the stationary phase of growth and plays decisive roles in cell adaptation to the new conditions and survival. A typical mechanism is the production of GASP (Growth Advantage in Stationary Phase) phenotypes [58,59,60], which usually appear at prolonged starvation. GASPs are generated as a result of mutations in the *rpoS* gene, which apparently confers advantages for growing under restrictive conditions and potentially outcompeting the parental population [60,62] by scavenging nutrients from lysed cells [59,71]. A number of bacteria have already been reported to enter GASP phenotypes and so these processes are expected to be generalized among prokaryotes [54]. This is an example of diversification that confers advantages in changing environments or under varying conditions [100,101,103]. Another phenomenon, mainly observed at the late stationary phase, is maintaining a working cellular machinery for protein synthesis, which has received the term CASP (Constant Activity at Stationary Phase) [116,117]. Among other potential aspects for future studies is the phenomenon named SCDI (Stationary phase Contact-Dependent Inhibition) [61,62], which inhibits the growth and activity of contacting cells to avoid competition. Stress-induced mutagenesis has been reported [53] as a consequence, for example, of nutrient deprivation based on stationary phase-specific processes. The molecular basis for these responses has been studied, although numerous questions remain to be answered [52,54].

During the stationary phase of growth, cells in a batch culture can be at a variety of growth/death stages. This phase of the growth curve is expected to present large variability in the status of bacteria within the population being cultured. A number of physiological phenomena have been mentioned above that result in cells at different physiological stages (see below). The stationary phase might represent a source to generate genetic diversity [52,53,54,57,58,103], the first steps of a speciation process in microorganisms. Nevertheless, confirmation and a vision for extending that variability remain to be understood.

### 6.2. Decline (Decay) Phase of the Growth Curve

At advanced stages of the stationary phase of growth, the decay phase is aimed at reducing the number of cells in the culture (Figure 1) so that some can survive and the population can achieve a much longer survival time at a cost of losing a large fraction of its members [54]. Herein, this phase is preferentially named the decline phase, although other names used for this period of the growth curve are the decay or death phase. The result in this period is a reduction in cell number in the culture or bacterial population, leading to the late stages of the stationary phase and cell reduction that result in an asymptotic shape in the final portion of the growth curve (Figure 1).

### 6.3. Late Stages of the Stationary Phase of Growth

The late stationary phase of growth (Figure 1) or long-term stationary phase [52,68,118], even beyond the decline phase, presents the most drastic strategies employed to survive the adverse conditions through long time periods in batch cultures. Usually the phenotypes and mechanisms described above apply in this phase. The aim is to persist in the environment and, to that aim, some adaptations are required so that the most well-fitted will persist. The usual shape of the curve at this point is clearly asymptotic and a number of different phenotypes are likely to appear. The time scale can change in time units throughout the growth curve; this is illustrated from hours (lag and exponential phases) up to months and years (late stages of the asymptotic decay curve) although the scale is species-dependent (Figure 1). Differences among cells might or might not be inherited so scientists could differentiate among genomic changes at this point between inheritable ones (generally described as polymorphisms, mutations, new gene incorporation, or deletion, etc.) [53,119] and epigenetic changes or differential expression responses (non-heritable) that assign differential functionalities to specific phenotypes [99,100,101,103]. Epigenetic differences have been reported to occur throughout the late stages of the stationary phase [99,100,101,103].

### 6.4. Microbial Decay and Survival

Microorganisms are often exposed to tough adverse conditions that induce mechanisms for survival. These conditions often result in a decay of cell numbers over time, which can follow different kinetics depending on the microbial species and the negative factors leading to the decrease in cell abundance.

Microorganisms have developed different strategies of survival to face either starvation, antimicrobial substances or toxic products, and adverse conditions. For example, Gram-positive bacteria and fungi have developed spores that are specific forms of resistance to persist or survive adverse conditions [120]. Spores are morphologically differential structures designed to resist adverse conditions, but they cannot be considered as active cells; rather, they present strong cellular envelopes to resist the adverse environment and protect the genome and minimum machinery to restart growth once appropriate conditions return [121,122]. Other microorganisms have not developed specific structures for survival, but they present a number of strategies to maximize survival during adverse conditions that asymptotically extend the length of the survival curve and present, for instance, in periods of dormancy. An example is Gram-negative bacteria [32,33,62]. Microorganisms lacking resistance structures are also known to persist through responses involving morphological changes, such as cell length increase due to inhibition of cell division, restructuring of whole-cell gene expression to repress common metabolic pathways, activation of secondary metabolites, and the consumption of different and specific substances as nutrient and energy sources (i.e., switch from preferential consumption of sugars to use of lipids and proteins, etc.) [32,54,62].

The term bacterial death, or dead bacteria, has been quite freely used in a number of publications and texts. Bacterial death is an issue that deserves further debate. This is because bacteria can be unable to grow on culture media but are able to eventually develop and, for instance, colonize a given environment or, for a pathogenic bacterium, cause disease [121,122,123]. At present, bacterial death should be defined to apply to those bacteria that have their cell integrity seriously compromised. This is the most conservative definition to provide certainty of death. Bacterial lysis, as an example, will result in microscopically visible morphological changes and the cell can be confirmed to have lost cell integrity. Current methods to classify live/dead microorganisms are generally based on evaluating if a membrane-impermeable dye penetrates into the cells [123,124]. Living cells will not be stained with the differential dye, but stained cells are those with a compromised cell envelope and the dye has been able to penetrate into the cell likely because of compromised cell membrane integrity [123]. Non-culturable cells should not be necessarily considered dead cells because they can present certain metabolic activity, they may be able to grow under different, more appropriate, conditions or develop in a suitable host initiating pathogenesis [122]. Similarly, dormant cells are not necessarily dead; rather, they are employing a survival strategy that will allow them to persist through adversity and regrow at a latter point in time when conditions improve [68,124,125].

### 6.5. Maintenance Metabolism

Cells require a minimum metabolism to maintain cell functioning. This minimum activity is named maintenance metabolism. Beyond this, additional nutrients can be directed to growth. Maintenance metabolism, as originally described [125,126,127], is usually defined as the metabolism required for non-growth (zero-growth condition)-related processes; it is a basal metabolism [68,127]. Microbial cells are often exposed to conditions that allow them to obtain sufficient energy sources only to cope with the requirements of maintaining minimum cell metabolism with no growth. Typical processes required for maintenance metabolism are those other than generation of new biomass. Examples are nucleic acid and protein repair, sustaining the proton motive force, minimum membrane potential, osmoregulation, etc. [27,68,107,118,126,127]. In theory, the maintenance metabolism of a microbial species is represented by the limit of the asymptotic growth rate curve during retentostat cultivation [87,88,89,90,104,107] (Figure 3). This can represent an approach for estimating maintenance metabolism and solving a long debate about strategies to estimate the energy required for those minimum basal conditions [127,128]. Models extracted from laboratory cultures have been used to approach preliminary estimates in the environment [68].

## 7. Physiological Stages and Methods

Classifying microorganisms at different physiological stages besides the growth phases described above has been tempting to differentiate cells in temporal and environmental situations. Generally, this is a step forward in understanding microbial behavior in the environment and the different scenarios for growth and decay. Description and use of microbial physiological states occurs in parallel with the methodology used to characterize those groups or phenotypes of cells. Through years, additional types of cells have been reported based on the methodology used and it has not been always easy to compare physiological stages using different methods.

The first physiological states defined for bacteria were related to culturability. Bacteria can be cultured or not. It is known that many bacterial taxa cannot be cultured yet (see above). Within a mono-specific culture, there is a fraction of cells that are not able to grow in culture and so they do not appear in colony counts. Thus, counts of CFU (Colony-Forming Units) to enumerate culturable cells are generally lower than the total cell counts enumerated by counting total cells under a microscope. These methods clearly quantify culturable (CFU) and non-culturable (difference between total and CFU counts) cells.

Bacteria are also described as viable. A viable cell is that one capable of actively thriving in its environment. A viable cell should show some kind of metabolic activity [64,121,124]. It can be cultured or not. Those viable cells that cannot be cultured have been classified as viable but non-culturable (VBNC) cells [108,120]. Viable cells have been determined using indicators of activity either by spectrometric measurements (for batch determinations) or microscopic counting (to determine active cell counts). Some methods used to this aim include the use of nalidixic acid (which elongates active bacteria and inhibits cell division) [129], measurements of membrane potential (to differentiate metabolically active and inactive cells) [130], FDA hydrolysis (fluorescein diacetate; a fluorogenic compound that fluoresces after enzymatic hydrolysis) [131], microautoradiography using radiolabeled substrates (glucose, thymine, amino acids, etc.) [132] or stable isotope labeling as in NanoSIMS [82,95], among multiple other procedures [96,124].

Active bacteria are those showing some metabolic activity [95,127,128,129], and they can be evaluated using some of the methods described in the previous paragraph, among others. One should assume that an active bacterium is a viable cell at least, while we do not have a practical method to determine which level of minimum cellular activity defines a viable state according to the bacterium’s maintenance metabolism.

The question about live vs. dead microbial cells has been of interest for a long time. Current common methods involve the use of differential staining. Those cells with compromised cell envelopes are stained by a cell-impermeable dye (e.g., propidium iodide) and the others can be visualized using a general cell staining dye (e.g., syto9). A common current methodology for live/dead counts uses a mixture of propidium iodine and syto9 fluorescent dyes to achieve differential staining [123]. As mentioned above, cells presenting compromised cell envelopes could be considered dead. Although some ambiguity is known to exist, until we determine the level of cell envelope compromise needed to realize cell death, this method is recommended and frequently used, apparently showing satisfactory results [123] for a variety (not all) of bacterial species.

The determination of dormant cells other than by subtraction of other stages from the total cell count becomes more complicated [71,124]. Dormant cells could be described as those that show very low or no level of activity during a specific time period (i.e., the incubation time performed to carry out the determination), but they are known to be able to start growing when provided adequate conditions and nutrients. Accordingly, dormant cells could be enumerated by colony count and thus they could be culturable cells. Dormant cells can show colony formation in culture even if they show little or no activity in other activity tests. The case of spores is similar [120]. Spores can show no activity, but they can be cultured under appropriate conditions and so they could be described as inactive but culturable cells.

## 8. Conclusions and Perspectives

Understanding microbial life is relevant because of the great importance of microorganisms in ruling environmental processes and our own survival. At present, it is of the utmost interest to be able to comprehend how microorganisms and microbial communities behave under a broad range of conditions and changing scenarios. Understanding the microbial world will allow us to appropriately manage and use microorganisms to sustain the environment as well as access microbiome functionality as it affects humans and other living creatures. At present, microbiology is a truly multidisciplinary science because microorganisms are critically involved in multiple scenarios, such as clinical and medical fields, human and animal health, plant growth and agriculture, environmental and biogeochemical issues, climate warming, biotechnology, waste processing, all green and sustainable processes, etc. (Table 1).

For microorganisms to persist in the environment and its changing conditions, they have developed different growth states. Each state shows specific peculiarities with a remarkable adaptive capacity to thrive under a variety of conditions. The most common and better understood restrictive environmental condition for microorganisms involves nutrient limitation. Thus, microorganisms have adapted to thrive under restricted conditions by using a number of strategies, some outlined above and some others yet to be described, that allow them to thrive through specific scenarios. The study of microorganisms under severely growth-limiting conditions represents a major scenario for the advancement of our understanding of the microbial role in the environment.

How microorganisms adjust their metabolism and cell machinery in response to environmental changes (e.g., nutrient availability, temperature, pH, salinity, inhibitors, etc.) brings important outstanding questions about bacterial physiology and both the intra- and inter-species variability of microorganisms [32]. This opens the door to pursuing critical research on the complexity of regulatory networks of gene expression and microbial adaptation. The adaptation of microorganisms to different environmental conditions, above all to stress situations (i.e., nutrient scarcity and adverse factors), is expected to generate new diversity. We hypothesize that this diversification develops at a much faster pace under stress factors than under ideal, optimum growth conditions. Then, studies under the most growth-limiting conditions are of primary importance for the advancement of current microbiological knowledge. Diversity and increase in diversity are aspects of remarkable interest for the future of environmental sustainability as well as sources for novel applications, for example, in biotechnological and clinical fields.

Future developments are needed to fully understand and monitor microbial life on Earth. Throughout the above text, some aspects requiring further research have been proposed for debate and open collaborations. Novel methodologies are needed [133] to rapidly and efficiently determine microbial growth status and physiological state. Understanding microbial responses to stress and limitations but lacking monitoring tools will result in truncated applicability and limitations in the management of the microbial components of ecosystems. Detailed knowledge of microbial functioning will allow us to interpret and manage environmental issues at a local level (e.g., ecosystems, their health, and sustainability) as well as at the global scale (e.g., climate warming).

## Figures and Tables

**Figure 1 microorganisms-11-01641-f001:**
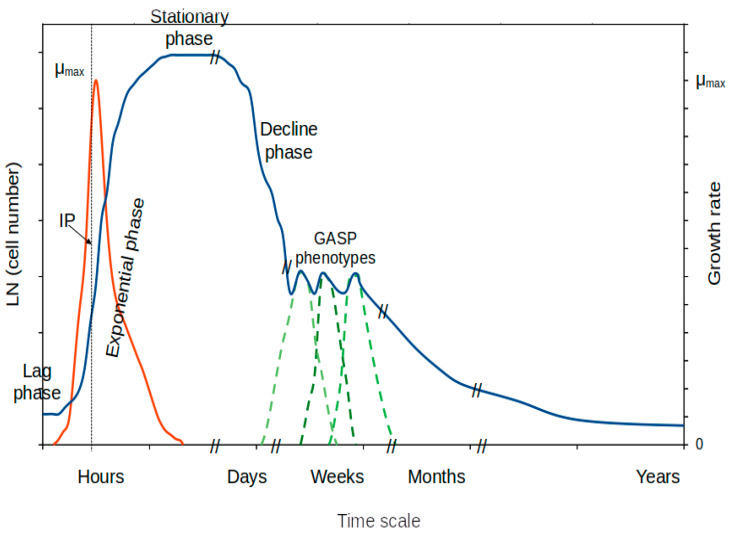
A typical growth curve for a mono-specific microbial culture in batch showing the distinctive phases of growth with incubation time. IP, inflection point; μ_max_, maximum growth rate. Abundance, blue line; growth rate, red line; dashed green lines, distinct GASP phenotypes.

**Figure 2 microorganisms-11-01641-f002:**
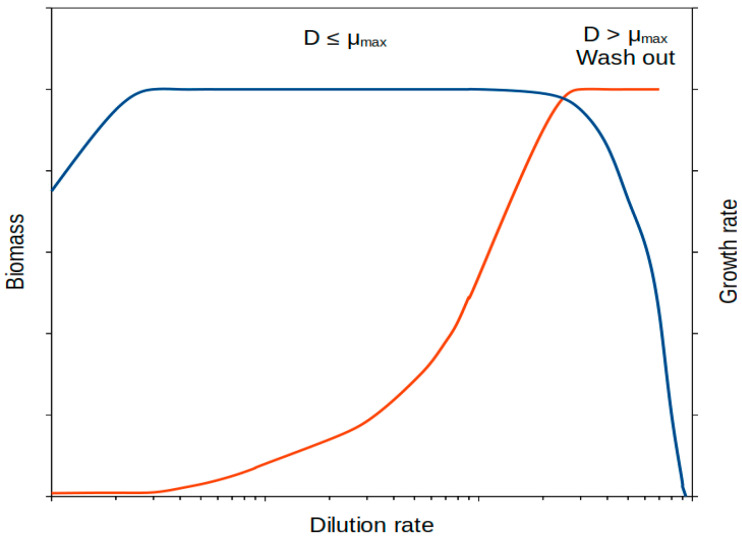
Growth rate and relevant parameters during a continuous culturing system in a chemostat. Dilution rate, in logarithmic scale, strictly determines growth rate in a chemostat. Beyond the point when the dilution rate exceeds the maximum growth rate, a wash out of cells occurs. Biomass, blue line; growth rate, red line.

**Figure 3 microorganisms-11-01641-f003:**
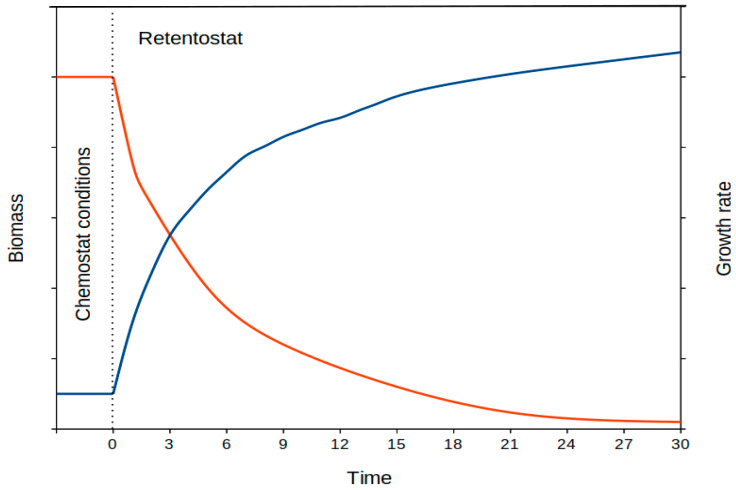
Representation of the biomass and growth rate over time in a retentostat culturing system (with complete biomass retention) for a model microorganism. The progressive increase in biomass (blue line) over time follows a logarithmic curve model, which is used to estimate the actual growth rate (red line) of the cells at a given time point.

**Table 1 microorganisms-11-01641-t001:** Characteristics of different growth states of microorganisms, distinctive growth rates, functional aspects of their study, the influence of the environment, and relevance to understanding microbial responses.

Characteristics /Relevance	Optimum Growth	Slow Growth	Near-Zero Growth	Stationary Phase	Maintenance Metabolism
Conditions	Optimum/ Laboratory	Restricted growth	Severely limited growth	Nutrient depletion/growth inhibition	Maintenance /Persistence
Expected net growth	Optimum/maximum	≥0.025 h^−1^	<0.025 h^−1^	0–Decline	0
Degree of understanding	High	Medium/Low	Low	Low	Low
Environmental relevance	Poor	High	High	high	high
Value in taxonomy *	High	Poor	Poor	Poor	Poor
Discovering cell capabilities	Limited	Enhanced	High	High	High
Expected variability	Low	High	Very high	Very high	Medium
Secondary metabolite discovery	Low	High	Very high	High	Medium
Genome understanding	Medium	High	High	High	High
New gene annotation	Low/Medium	Medium/High	High	High	High
Microbial behavior understanding	Basic	Medium	High	High	High
Microbiome analysis value	Poor	Medium	High	High	Medium
Understanding adversity	Null	Medium	High	High	High
Microbial interaction potential	Poor	Medium	High	High	High

* According to present requirements.

## Data Availability

No new data were created or analyzed in this study. Data sharing is not applicable to this article.

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
