# Peer review of "Microbial Growth under Limiting Conditions-Future Perspectives"

_microorganisms, 2023, doi:10.3390/microorganisms11071641_

Round 1

Reviewer 1 Report

1. Although a vast amount of data has been compiled, the necessity and purpose of this review paper should be presented in the introduction

2. It is not clear what the author intends to say through this manuscript, and what conclusions were drawn from this review paper?

3. It is unclear that criteria the table of contents was decided on, and some chapters are too long and some are too short. The whole contents needs to be reorganized and rearranged.

Author Response

Reviewer # 1

Comments and Suggestions for Authors

1. Although a vast amount of data has been compiled, the necessity and purpose of this review paper should be presented in the introduction

Response:

We have improved the ms following the Reviewers comments in several ways (see below). About the necessity of indicating the purpose of the ms, we have added in the introduction a sentence on this (lines 157-160):

This study aims to highlight the interest and relevance of studying microbial behavior under growth-limiting conditions as the next, and necessary, step towards better understand microbial life and its role in the environment.

Besides, throughout the ms and in the last section (Conclusions and perspectives) we have indicated the purposes of this ms in relationship to novel aspects that require further research.

We are attaching to this submission a copy of the track changes introduced in the ms to facilitate a follow up of the changes.

2. It is not clear what the author intends to say through this manuscript, and what conclusions were drawn from this review paper?

Response:

Major lines of conclusions from this manuscript is to highlight the requirements for further research on growth-limited microorganisms as a first step to be able to understand a large number of issues related to bacterial physiology, behavior under a variety of environmental conditions, the variability within populations and complex communities in relationship to growth and activity, and the potential future perspectives to be able to manage microbial communities both in the environment and as part of human, animals and plant microbiomes.

This has been more clearly highlighted in the Conclusions section as well as throughout the whole ms. To this aim, also the reduction of length in some sections of the ms will certainly assist to better understand the purposes and conclusions to be extracted from this manuscript.

3. It is unclear that criteria the table of contents was decided on, and some chapters are too long and some are too short. The whole contents needs to be reorganized and rearranged.

Response:

We would like to offer a perfectly metric manuscript where all sections are equal in length but this is not the case because some sections require longer text than others in order to clearly present the different issues. We have shortened some of the longest sections which will certainly to reduce the effects that the Reviewer has observed. We appreciate your interest in improving the ms and, be sure, that we have tried our best to comply with your suggestions.

Reviewer 2 Report

This is an interesting discursive synthesis of knowledge of and research methods used  in basic cell microbiology. It points out the need for considerable development in methods used to study microbial adaptation and growth, without going into details. So it is the concept behind the article that is important.

First, I would like to suggest that the tile be changed to become more descriptive and attractive. Perhaps "Microbial growth under limiting conditions: future perspectives", or "Near-zero growth of microbial cells: a necessary study for future developments?" 

Secondly, it is essential that the article be thoroughly checked by a native English speaker. There are many errors.

The authors should consider for whom they are writing this article. Much of the text is information well-known to all microbiologists. For example, it is not necessary to describe the stages of the bacterial growth cycle. The text can be reduced drastically in length. In my opinion, it is much too long.

More detailed comments:

lines 31, 34 and 52 - replace i.e. with e.g.

line 59 - isn't rather growth a result of activity than the reverse?

line 70 - ref no. 29 is too old (1983)e used here

line 165-6 - is this gap really expected to be filled in the NEAR future? Maybe a reference is required for this?

line 215 - remove "among others"

In Section 3, I think it would be interesting to actually point out that the growth curve is basically a laboratory phenomenon, with a single culture in a single (generally man-made) medium and that it cannot be assumed that growth conditions and cell/genetic adaptations found here actually occur in the real environment.

In Section 4, the authors seem to be assuming that all slow-growers are so because of the sub-optimum conditions. Do you not consider that some may simply not have the machinery to grow faster?

lines 581-2 - It is not clear whether contact with other cells, or adhesion to a surface is meant. Please clarify.

The final section should be titled "Conclusions and Future......"

As stated above, it is essential that the English be corrected. Spelling is good, but grammar, punctuation and phraseology are poor.

Author Response

Detailed response to Reviewers’ comments

Reviewer #2

Comments and Suggestions for Authors

This is an interesting discursive synthesis of knowledge of and research methods used in basic cell microbiology. It points out the need for considerable development in methods used to study microbial adaptation and growth, without going into details. So it is the concept behind the article that is important.

First, I would like to suggest that the title be changed to become more descriptive and attractive. Perhaps "Microbial growth under limiting conditions: future perspectives", or "Near-zero growth of microbial cells: a necessary study for future developments?"

Response:

Thank you very much for your assessment of the ms and the time used to carefully read and comment it.

Following your suggestion, we have changed the title as you proposed, and we agree that the corrected title is better than the previous one. The improved title is: “Microbial growth under limiting conditions. Future perspectives”

Secondly, it is essential that the article be thoroughly checked by a native English speaker. There are many errors.

Response:

The ms has been thoroughly checked introducing many corrections as the Reviewer indicates. This is a major improvement to understand its content. The previous ms with track changes is submitted to easily follow all these changes.

The authors should consider for whom they are writing this article. Much of the text is information well-known to all microbiologists. For example, it is not necessary to describe the stages of the bacterial growth cycle. The text can be reduced drastically in length. In my opinion, it is much too long.

Response:

We agree with your observation and the section containing the description of the growth curve has been drastically reduced (about 20 lines shorter). We believe that the figure is important to visualize some of the aspects indicated in the text and so, we consider that Figure 1 should be kept in the manuscript although the description for this figure (as mentioned) has been simplified to lines 165-169. Figure 1 assists to understand the estimations of growth rates and to the non-specialist it will help to get introduced in the topic. The ms is mostly aimed to researchers focusing on understanding bacterial behavior and we point out a number of requirements needed to fill several of the present gaps in our knowledge on bacterial growth and activity. Above all, we highlight the needs to understand how bacteria respond to growth limitations and the stages when their growth is severely limited as is believed that microorganisms are most frequently in nature. Different methodological aspects are indicated, as well as some points in need of further research. The understanding of microbial growth rates and their response to environmental factors is a fist required step to comprehend how bacteria behave and consequently how to use and manage their populations and the complex microbial communities.

More detailed comments:

lines 31, 34 and 52 - replace i.e. with e.g.

Response: Corrected as suggested. Now in lines 33, 36 and 52. Also corrected in lines 744 and 745.

line 59 - isn't rather growth a result of activity than the reverse?

Response: Yes, you are right, we agree. We have corrected the sentence (now line 60): Growth is a result of microbial activity

line 70 - ref no. 29 is too old (1983)e used here

Response: We have removed previous reference 29 and introduced two recent references: 30 and 31, corresponding to Himeoka & Mitarai (2020) and Zhu & Dai (2023).

line 165-6 - is this gap really expected to be filled in the NEAR future? Maybe a reference is required for this?

Response: This paragraph has been corrected introducing references to justify and to clarify the content. This is now in lines 152-160. The gaps are required for further advancement on understanding microbial responses to environmental factors so that potential uses and consequences of microbial activity and growth and the management of microbial communities can be envisioned and assessed.

line 215 - remove "among others"

Response: Removed as indicated. This sentence corresponds to the line 190 in the corrected manuscript.

In Section 3, I think it would be interesting to actually point out that the growth curve is basically a laboratory phenomenon, with a single culture in a single (generally man-made) medium and that it cannot be assumed that growth conditions and cell/genetic adaptations found here actually occur in the real environment.

Response: We have indicated several additional times (throughout the whole manuscript) that the growth and the growth curves correspond to bacteria in the laboratory and under laboratory conditions. Laboratory gorwth of microorganisms is always a tool to start understanding microorganisms in a simplified environment but it is also a required first and necessary step towards the goal of understanding microbial behavior in nature.

Examples where we have indicated this point in the corrected version of the manuscript:

Lines 63, 163, 164, 173, 178, 191, 193, 227, 284, 286, 288, 295, 313, 334, 416, 435, 645.

In Section 4, the authors seem to be assuming that all slow-growers are so because of the sub-optimum conditions. Do you not consider that some may simply not have the machinery to grow faster?

Response: This has been considered. A sentence indicating this point has been added in lines 117-118 and in lines 223-226. We indicate that some microorganisms are fast growers and others are not and that the terms slow growth rate is relative to the microorganism optimum (maximum) growth rate.

lines 581-2 - It is not clear whether contact with other cells, or adhesion to a surface is meant. Please clarify.

Response: The phenomenon of SCDI (Stationary phase Contact-Dependent Inhibition) has been reported by several authors. Contact between appears to be required independently of the existence of a surface to adhere or biofilm formation. For further and detailed information I recommend the paper by Lemonnier et al. (reference Num. 61; The evolution of contact-dependent inhibition in non-growing populations of Escherichia coli. Proc Biol Sci 2008, 275, 3-10. doi: 10.1098/rspb.2007.1234). We believe it is not the point to expand excessively on this besides pointing out this phenomenon.

The final section should be titled "Conclusions and Future...…"

Response: The final section has been revised and the title for this section has been changed following the Reviewer’s suggestion to: “Conclusions and perspectives”

Comments on the Quality of English Language

As stated above, it is essential that the English be corrected. Spelling is good, but grammar, punctuation and phraseology are poor.

Response: As commented above, we have thoroughly checked the ms to correct the sentences and grammar. This has been mostly a consequence of attempting to meet a deadline for the compromise of submitting this manuscript. We apologize about this.

Round 2

Reviewer 1 Report

No special comments